# MHD Double-Diffusive Carreau Fluid Flow through a Porous Medium with Variable Thermal Conductivity and Suction/Injection

**DOI:** 10.3390/e24030377

**Published:** 2022-03-08

**Authors:** Salman Zeb, Shafiq Ahmad, Muhammad Ibrahim, Tareq Saeed

**Affiliations:** 1Department of Mathematics, University of Malakand, Chakdara 18800, Dir (Lower), Khyber Pakhtunkhwa, Pakistan; salmanzeb@uom.edu.pk (S.Z.); shafiq49043@gmail.com (S.A.); 2Department of Basic Sciences, CECOS University of IT and Emerging Sciences, Peshawar 25000, Pakistan; 3Nonlinear Analysis and Applied Mathematics (NAAM)-Research Group, Department of Mathematics, Faculty of Science, King Abdulaziz University, P.O. Box 80203, Jeddah 21589, Saudi Arabia; tsalmalki@kau.edu.sa

**Keywords:** variable thermal conductivity, Carreau fluid, double diffusion, magnetic field, porous medium, suction and injection

## Abstract

In this article, we consider the effects of double diffusion on magnetohydrodynamics (MHD) Carreau fluid flow through a porous medium along a stretching sheet. Variable thermal conductivity and suction/injection parameter effects are also taken into the consideration. Similarity transformations are utilized to transform the equations governing the Carreau fluid flow model to dimensionless non-linear ordinary differential equations. Maple software is utilized for the numerical solution. These solutions are then presented through graphs. The velocity, concentration, temperature profile, skin friction coefficient, and the Nusselt and Sherwood numbers under the impact of different parameters are studied. The fluid flow is analyzed for both suction and injection cases. From the analysis carried out, it is observed that the velocity profile reduces by increasing the porosity parameter while it enhances both the temperature and concentration profile. The temperature field enhances with increasing the variable thermal conductivity and the Nusselt number exhibits opposite behavior.

## 1. Introduction

Non-Newtonian fluids have gained great attention due to their immense applications in industrial and biomedical applications. The study concerning electrically conducting non-Newtonian fluids passing through a stretching sheet has been done by many researchers due to its vast engineering and industrial applications comprising plastic and metal extrusion, insulating materials, metal spinning, and in insulating materials. Sahoo [1] studied the non-Newtonian fluids past a stretched sheet while considering partial slip. It was found that the slip diminishes the momentum boundary layer thickness while enhancing the thermal boundary layer. Akbar et al. [2] explained the magnetohydrodynamics (MHD) flow of the tangent hyperbolic fluid through a stretching surface. They concluded that the velocity exhibits a decreasing behavior due to augmentation in magnetic parameter, power index, and the Weinssenberg number. Hamid et al. [3] found the dual solution and performed an analysis of heat transport and flow of the MHD Casson fluid towards an expanding sheet. The results showed that the fluid gives steady profiles for positive eigenvalues. Sharma and Bisht [4] studied MHD Sisko nanofluid along with joule heating through a stretching sheet. The results revealed that with joule heating, the velocity profile reduces whereas the temperature increases significantly. Salahuddin et al. [5] studied the MHD flux of the Williamson fluid towards a stretching sheet under the Cattenneo–Christov heat flow model with varying thickness. Their results showed that the impact of the magnetic parameter is opposite on the velocity whereas temperature enhances due to an increase in the magnetic parameter. Additionally, greater values for the wall thickness were considered suitable for the reduction in the velocity profile. Zakir et al. [6] studied MHD tangent hyperbolic fluid slip flow through stretching sheet. For further study related to fluid flow problems through a stretching sheet we refer to [7,8,9,10,11,12].

Double diffusion in fluid flow is a phenomenon that discusses a type of convection with two different density gradients both having different diffusive rates. This is a very important phenomenon and many researchers are driven to study it. Malik et al. [13] analyzed Sisko fluid under the Cattenneo–Christov double-diffusive model. It is deducted that with great relaxation times the temperature, as well as the concentration profiles, reduce. Waqas et al. [14] studied a chemically reactive non-Newtonian fluid with improved double diffusion. The concentration profile increases more for the destructive chemical reaction parameter than the generative parameter. Haq et al. [15] discussed stagnation point flux with the magnetic field, thermal radiation, and slip effect towards an expanding sheet and the results revealed that temperature elevates with increasing thermophoresis parameter while, due to the Brownian motion, the concentration profile reduces. Shankaralingappa [16] discussed double-diffusive Oldroyd-B fluid flow along a stretch sheet while utilizing Cattaneo–Christov heat theory in addition to the consideration of thermophoretic particle deposition, heat source/sink, and relaxation chemical reaction. Their results showed that velocity profile declines for rotation parameter and temperature and concentration profiles decline for increasing relaxation time parameter, and the concentration distribution reduces for enhancing values of chemical reaction rate and thermophoretic parameter. Kumar et al. [17] analyzed viscous ferromagnetic liquid flow along a stretched cylinder having thermophoretic particle deposition and with a uniform heat source/sink. Their outcomes described that the velocity field reduces when enhancing the thermophoretic coefficient and parameter, and the thermal gradient enhances when raising the ferromagnetic interaction parameter, while the opposite behavior is seen for the heat source/sink parameter.

Fluid flow via a porous medium has a significant amount of applications in petroleum engineering, geothermal and industrial operations. A porous medium is a solid matrix having a continuous network of pores. A porous medium which allows fluid flow can be natural including sand, wood, and human lungs or it can be synthetic such as ceramics, metal foams having high porosity, and composite materials. Pores and interconnected solid particles make up porous media which can be seen in electrochemical systems, steel and iron production, microchemical reactors, and renewable fuels. Traditionally, Darcy’s law is utilized mainly for computational and theoretical investigations of porous medium studies. Heat transfer effects and fluid flow through a porous medium are studied by numerous researchers. Khan et al. [18] studied triple diffusive flow saturated by nanofluid through a permeable horizontal plate and showed that heat transfer rate enhances with adding nano-particles and salts. Krishna and Reddy [19] discussed the MHD non-Newtonian forced convective flow via porous medium through a stumpy and deduced that heat transfer is dominated at low porosity by conduction while for high porosity it is dominated by convection. Hayat et al. [20] carried out the analysis on MHD slip flux and heat transport effects over a penetrable stretching sheet. Siavashi and Rostami [21] studied non-Newtonian nanofluid with natural convective properties along a concentric circular region with completely or partially filled porous medium. They considered the two phase simulations of the fluid and the results disclosed that a fully porous cavity generates less entropy because in the porous zone the fluid has the lowest temperature gradients. Eldabe et al. [22] discussed the viscous dissolution effect on the free convection flux via porous media.

The Carreau fluid flow model is a significant non-Newtonian fluid model. It effectively describes both shear thickening and shear thinning phenomena. At a high wall shear stress, the behavior of this fluid degenerates to a Newtonian behavior. This model considerately explains the rheological behavior of many industrial fluids such as foams, cosmetics, syrups, and biotechnological detergents. Khan et al. [23] discussed the effects of heat transfer of squeezed Carreau fluid passing through a sensor area with changing thermal conductivity. Velocity increases with both enlarging the permeable velocity and the squeezed flow parameter. A study of the heat and mass transport impacts of radiative Carreau fluid with a magnetic field is presented by Machireddy and Naramgari [24]. Sulochana et al. [25] explained the transpiration impact of Carreau nanofluid with stagnation-point flow amid Brownian motion and thermophoresis. The study explains that the heat and mass transport rates elevate with the thermophoresis parameter. The impact of solar radiation and heat generation on Carreau nanofluid with varying thickness passing across a stretched sheet is studied by Khan et al. [26]. Velocity enhances with wall thickness parameter, and heat generation and radiative heat parameter cause an increase in temperature. Raju and Sandeep [27] studied MHD Carreau fluid across a wedge having the effects of cross-diffusion and found that decelerating flow past a wedge is good for cooling. Non-Newtonian MHD Carreau fluid in a sphere is investigated numerically by Amanulla et al. [28]. Akbar et al. [29] studied peristaltic flux of Carreau nanofluid through an uneven channel. Pressure rise enhances with the magnetic and thermophoresis parameters. The authors in [30] investigated MHD Carreau fluid flow along a variable stretch sheet and found that velocity field declines for enhancing power-law index and magnetic parameter values.

In this article, we examine MHD Carreau fluid flow in permeable media over a stretched sheet. Double diffusion, variable thermal conductivity, and Darcy’s law are employed for porous medium in the modeling of the problem. Equations governing the considered Carreau fluid flow model problem are transformed to non-linear ordinary differential equations in dimensionless form by employing similarity transformations. The effects of pertinent parameters present in the problem for injection/suction cases on velocity, temperature, concentration, and on other physical quantities such as skin friction and Nusselt and Sherwood numbers are investigated, represented graphically, and discussed accordingly.

The remaining article is organized as follows: The problem formulation is presented in Section 2. In Section 3, we analyzed our results, while the conclusion of the work is given in Section 4.

## 2. Problem Formulation

We consider steady incompressible flow of a double diffusive Carreau fluid through porous medium over a stretching sheet in two dimensions. Assuming that the sheet is considered in the *x* direction with stretching velocity $u_{w}=ax$, where a>0 is a constant and is the stretching rate. The flow is restricted to the area y≥0 and Darcy’s law is applied for porous medium. A transverse uniform magnetic force field $B_{0}$ is inflicted across the *y*-axis. The geometry of the considered flow phenomena is depicted in Figure 1. The governing equations for the fluid flow model are formulated as [24,30,31].
(1)$$\frac{\partial u}{\partial x}+\frac{\partial v}{\partial y}=0,$$
(2)$$u\frac{\partial u}{\partial x}+v\frac{\partial u}{\partial y}=\nu \left(\frac{\partial^{2}u}{\partial y^{2}}+\frac{3(n-1)}{2}\Gamma^{2}\frac{\partial^{2}u}{\partial y^{2}}{\left(\frac{\partial u}{\partial y}\right)}^{2}\right)-u\left(\frac{\sigma B_{0}^{2}}{\rho }+\frac{\nu }{K_{1}}\right),$$
(3)$$u\frac{\partial T}{\partial x}+v\frac{\partial T}{\partial y}=\frac{1}{\rho c_{p}}\left(\frac{\partial }{\partial y}\left(k\left(T\right)\frac{\partial T}{\partial y}\right)\right)+D_{TC}\frac{\partial^{2}C}{\partial y^{2}},$$
(4)$$u\frac{\partial C}{\partial x}+v\frac{\partial C}{\partial y}=D_{SM}\frac{\partial^{2}C}{\partial y^{2}}+D_{CT}\frac{\partial^{2}T}{\partial y^{2}}.$$

The corresponding conditions on the boundaries are as follows
(5)$$\begin{array}{cc} u=u_{w} & =ax,\;v=v_{w},\;T=T_{w},\;C=C_{w}\;\mathrm{at}\;y=0,\\ \; & u\to 0,\;T\to T_{\infty },\;C\to C_{\infty }\;\mathrm{as}\;y\to \infty . \end{array}$$

In the above model, (Equation 1) is the continuity equation, (Equation 2) is the momentum equation, (Equation 3) is the energy equation, and (Equation 4) is the concentration equation. The terms *u*, *v* represent components of velocity along *x* and *y* directions respectively, ρ, σ, and ν represents density, electrical conductivity, and the kinematic viscosity of the considered fluid respectively, $D_{CT}$ is Soret type diffusivity, $D_{SM}$ is diffusivity of porous medium, $D_{TC}$ is Dufour type diffusivity, and $B_{0}$ denotes the applied magnetic force field while $c_{p}$ is specific heat and $K_{1}$ is porous medium permeability. Moreover, $u_{w}$ is stretching velocity of the sheet, $v_{w}$ is mass transfer velocity, *T*, $T_{w}$, and $T_{\infty }$ represent, respectively, the fluid temperature, temperature of the wall, and the ambient temperature, *C*, $C_{w}$, and $C_{\infty }$ are, respectively, the concentration, concentration at the wall, and free stream concentration, Γ is the material time constant, *n* is the power law index describing fluid characteristics, while $k\left(T\right)=k^{*}\left(1+\beta \left(\frac{T-T_{\infty }}{T_{w}-T_{\infty }}\right)\right)$ is the variable thermal conductivity.

The following similarity transformations are considered to transform the governing equations
(6)$$\begin{array}{cc} \; & \eta =\sqrt{\frac{a}{\nu }}y,\;\psi =\sqrt{a\nu }xf\left(\eta \right),\\ \; & \theta =\frac{T-T_{\infty }}{T_{w}-T_{\infty }},\;\phi =\frac{C-C_{\infty }}{C_{w}-C_{\infty }}. \end{array}$$

Subject to these similarity transformations, Equation (Equation 1) is automatically satisfied whereas Equations (Equation 2)–(Equation 4) with the boundary conditions (Equation 5) are transformed to the dimensionless form as
(7)$$f^{\prime }{}^{\prime \prime }-{\left(f^{\prime }\right)}^{2}+ff^{\prime \prime }+\frac{3(n-1)}{2}We^{2}{\left(f^{\prime \prime }\right)}^{2}f^{\prime }{}^{\prime \prime }-\left(M^{2}+\lambda \right)f^{\prime }=0,$$
(8)$$\theta^{\prime \prime }+\beta \left(\theta \theta^{\prime \prime }+{\left(\theta^{\prime }\right)}^{2}\right)+Prf\theta^{\prime }+P_{r}N_{d}\phi^{\prime \prime }=0,$$
(9)$$\phi^{\prime \prime }+Le\left(f\phi^{\prime }\right)+L_{d}\theta^{\prime \prime }=0,$$
(10)$$\begin{array}{cc} f^{\prime }\left(\eta \right) & =1,\;f(\eta )=S,\;\theta (\eta )=1,\;\phi (\eta )=1\;\mathrm{at}\;\eta =0,\\ \; & f^{\prime }\left(\eta \right)\to 0,\;\theta \left(\eta \right)\to 0,\;\phi \left(\eta \right)\to 0\;\mathrm{as}\;\eta \to \infty . \end{array}$$

Equations (Equation 7)–(Equation 9) are the transformed equations subject to the boundary conditions (Equation 10). In these equations, *n* is called the power law index, $We^{2}=\frac{\Gamma^{2}a^{3}x^{2}}{\nu }$ represents Weissenberg number, $\lambda =\frac{\nu }{aK_{1}}$ is porosity parameter, $M^{2}=\frac{\sigma B_{0}^{2}}{\rho a}$ is Hartmann number, $Pr=\frac{\mu c_{p}}{k^{*}}$ and $N_{d}=\frac{D_{TC}\left(C_{w}-C_{\infty }\right)}{\nu (T_{w}-T_{\infty })}$ represent the Prandtl number and modified Dufour parameter, respectively, $Le=\frac{\nu }{D_{SM}}$ is the Lewis number, $L_{d}=\frac{D_{CT}\left(T_{w}-T_{\infty }\right)}{D_{SM}\left(C_{w}-C_{\infty }\right)}$ is the Dufour solutal Lewis number, $S=\frac{-v_{w}}{\sqrt{a\nu }}$ where S>0 is for suction while S<0 corresponds to injection parameter.

The physical quantities such as skin friction, the Nusselt number and Sherwood number are given as
(11)$$C_{f}=\frac{\tau_{w}}{\rho {\left(ax\right)}^{2}},\;Nu_{x}=\frac{xq_{w}}{k(T_{w}-T_{\infty })},\;Sh_{x}=\frac{xh_{m}}{D_{SM}\left(C_{w}-C_{\infty }\right)}.$$

Here $\tau_{w}$, $q_{w}$ and $h_{m}$ are given as
(12)$$\tau_{w}=\mu \left(\frac{\partial u}{\partial y}+\frac{\left(n-1\right)}{2}\;\Gamma^{2}{\left(\frac{\partial u}{\partial y}\right)}^{3}\right),\;q_{w}=-k\frac{\partial T}{\partial y},\;h_{m}=-D_{SM}\frac{\partial C}{\partial y}.$$

Using (Equation 6) and (Equation 12), these quantities in transformed form are
(13)$$\begin{array}{cc} \; & \sqrt{Re_{x}}C_{f}=\left(f^{\prime \prime }\left(0\right)+\frac{\left(n-1\right)}{2}We^{2}{\left(f^{\prime \prime }\left(0\right)\right)}^{3}\right),\\ \; & {\left(Re_{x}\right)}^{-1/2}\;Nu_{x}=-\theta^{\prime }\left(0\right)\\ \; & {\left(Re_{x}\right)}^{-1/2}\;Sh_{x}=-\phi^{\prime }\left(0\right). \end{array}$$
where $Re_{x}=\frac{ax^{2}}{\nu }$ is local Reynolds number.

## 3. Results and Discussion

The non-linear transformed ODEs (Equation 7)–(Equation 9) along with boundary conditions (Equation 10) are solved numerically with the help of Maple software. The results of the effects of different parameters on the velocity profile ($f^{\prime }\left(\eta \right)$) and temperature (θ(η)) and concentration (ϕ(η)) profiles are presented graphically. We also analyzed the skin friction factor, Nusselt and the Sherwood numbers with regards to various parameters in detail. These parameters include *M*, the Hartmann number, We, the Weissenberg number, *n*, the power index, λ, the porosity parameter, Pr, the Prandtl number, β, variable thermal conductivity, Nd, the modified Dufour parameter, Le, the Lewis number, and Ld, the Dufour solutal Lewis number. In each graphical result, two cases are considered, one for S>0 which denotes suction with solid line graphs while the other S<0 is for the injection case with dashed line graphs. Skin friction coefficient values for different Hartmann numbers are compared in Table 1 with the work presented by Khan et al. [30] for analyzing the accuracy of our results which are showing close agreement with each other.

Figure 2a–d shows the changes in the dimensionless velocity profile $f^{\prime }\left(\eta \right)$ with reference to *M*, the Hartmann number, We, the Weissenberg number, λ, the porosity parameter, and *n*, the power-law index. In Figure 2a, we can see that raising values for *M* causes a reduction in velocity for both cases of suction and injection. As *M* corresponds to Lorentz force due to which for larger values of *M* the Lorentz force enhances and as this force is a resistive force acting against the motion of the fluid hence reduces the fluid’s velocity. Figure 2b illustrates the increasing behavior of the velocity field for We and for n>1. The velocity profile decelerates with larger porosity parameter (λ) as demonstrated in Figure 2c. This is because λ creates more resistance in the fluid due to Darcian drag. Additionally, the changes in velocity are greater in the case of injection as compared to the suction case. The effect of *n* is displayed in Figure 2d. It is evidently clear from this figure that the velocity enhances when *n* is increasing. This is because the non-linearity of the sheet is increased with the power-law index which reduces the resistant force and also the non-Newtonian behavior of the fluid is decreased.

Variations in temperature profile (θ(η)) of the fluid flow for various parameters are shown in Figure 3. In Figure 3a θ(η) is plotted against the variable thermal conductivity (β) and shows that temperature increases with higher thermal conductivity. This behavior is observed because the higher the thermal conductivity the higher will be the kinetic energy of the fluid particles which then elevates the temperature. Figure 3b demonstrates the effect of λ on θ(η). It is observed that θ(η) is enhancing due to the increase of the porosity parameter λ. It can also be seen that in the case of injection, the porosity parameter λ causes more changes in the temperature. The effects of Prandtl number (Pr) on the temperature field is plotted in Figure 3c. The reason for this particular behavior is that the Prandtl number has an inverse relation with the thermal diffusivity of the fluid so a higher Pr number means low thermal diffusion which then further means a lower temperature is observed. Figure 3d shows the impact of the modified Dufour parameter (Nd) on θ(η). It is clear that temperature increases with increasing Nd. In Figure 3e,f, the temperature profile against *M* and power-law index (*n*) are illustrated. It is deduced that θ(η) enhances with increasing *M* and reduces with *n*. Due to the greater magnitude of the Lorentz force depicted by higher *M*, the resistive force between the layers of the fluid results in an increase in the temperature.

In Figure 4a–d the dimensionless concentration profile is plotted against the Hartmann number (*M*), Dufour solutal Lewis number (Ld), Lewis number (Le), and porosity parameter (λ). From Figure 4a–d, it is clear that concentration profile (ϕ(η)) and the boundary layer thickness increases with increasing *M*, Ld, and λ while decreases with Le. The ratio of thermal diffusivity to the mass diffusivity is known as Lewis number due to which at high Lewis number thermal diffusion dominates, and that is why the concentration profile reduces. Due to porosity parameter λ, mass diffusivity is greater because of which ϕ(η) increases and λ has more effect on concentration in injection than in the suction case.

Figure 5a,b demonstrates behavior of the skin friction drag against variant values for *M*, *n* and λ. From Figure 5a, it is deduced that friction increases with increasing Hartmann number (*M*) and λ. An increase in *M* produces more friction between layers and also with the surface of the sheet. The opposite impact of the power-law index (*n*) on the skin friction coefficient is observed in Figure 5b.

The behavior of the local Nusselt number versus different parameters is observed in Figure 6a,b. Figure 6a shows that Nusselt number is decreasing for the variable thermal conductivity (β) plotted against porosity parameter (λ). As thermal conductivity increases the transfer of heat through conduction increases which results in a decrease in the Nusselt number. The behavior for the various values of the Pr versus Nd can be seen in Figure 6b. Increasing values of Pr means that momentum diffusivity dominates which results in more heat transfer and so the Nusselt number enhances.

Figure 7a demonstrates the impact of the Dufour solutal Lewis number (Ld) against λ on the Sherwood number. The results show that the Sherwood number lessens with both Ld and λ. The effect of the Lewis number (Le) versus the modified Dufour parameter (Nd) on the local Sherwood number is observed in Figure 7b where it is seen that the Sherwood number elevates with increasing values of Le and Nd.

## 4. Conclusions

In this article we studied a double-diffusive Carreau non-Newtonian fluid flow with variable thermal conductivity in a porous media. Similarity transformations are used to transform the governing Carreau fluid flow model equations into a system of non-linear ordinary differential equations. Physical quantities of primary interest are investigated for key parameters present in the study and a skin friction coefficient comparison has been carried out with the results available in [30] for various values of the Hartmann number which show agreement with each other. The impacts of various physical parameters are presented graphically. The following results are obtained from the present study.

The velocity profile decelerates for increasing Hartmann number (*M*) and porosity parameter (λ) while increases for Weissenberg number (We) and power-law index (*n*).The temperature field increases with variable thermal conductivity (β), λ, *M*, and with the modified Dufour parameter (Nd) while decreases for the Prandtl number (Pr) and *n*.The concentration profile reduces with the Lewis number (Le) while enhances with *M*, the Dufour solutal Lewis number Ld, and for λ.The local skin friction drag increases with *M* and λ while reduces with *n*.The Nusselt number diminishes with β, λ, and Nd while increases with Pr.The Sherwood number decreases with Ld and λ while increases with Le and Nd.The presence of the suction parameter provides more resistance to the fluid flow as compared to injection.The yemperature and concentration of a fluid increase in the case of injection parameter.

## Figures and Tables

**Figure 1 entropy-24-00377-f001:**
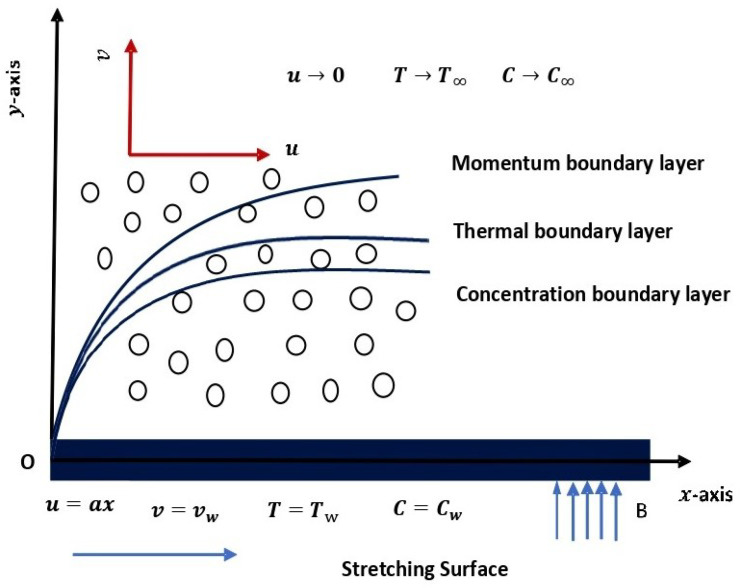
Flow geometry.

**Figure 2 entropy-24-00377-f002:**
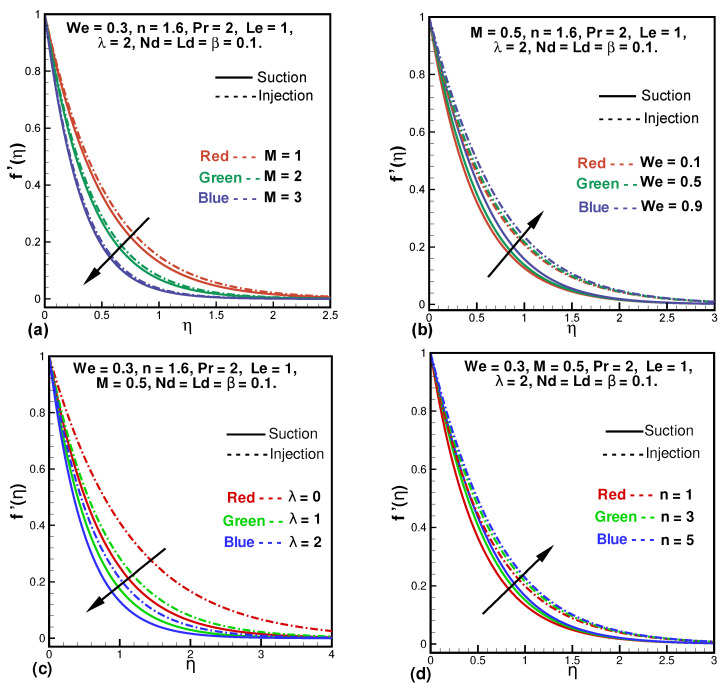
Variations in $f^{\prime }\left(\eta \right)$ against *M* (**a**), We (**b**), λ (**c**), and *n* (**d**).

**Figure 3 entropy-24-00377-f003:**
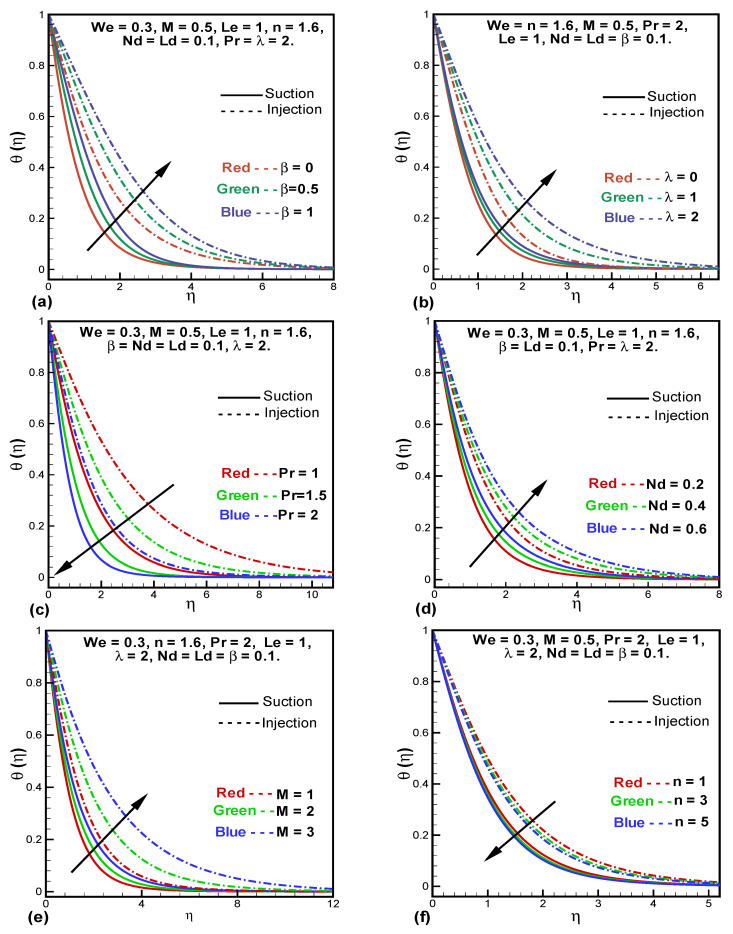
Variations in temperature profile against β (**a**), λ (**b**), Pr (**c**), Nd (**d**), *M* (**e**), and *n* (**f**).

**Figure 4 entropy-24-00377-f004:**
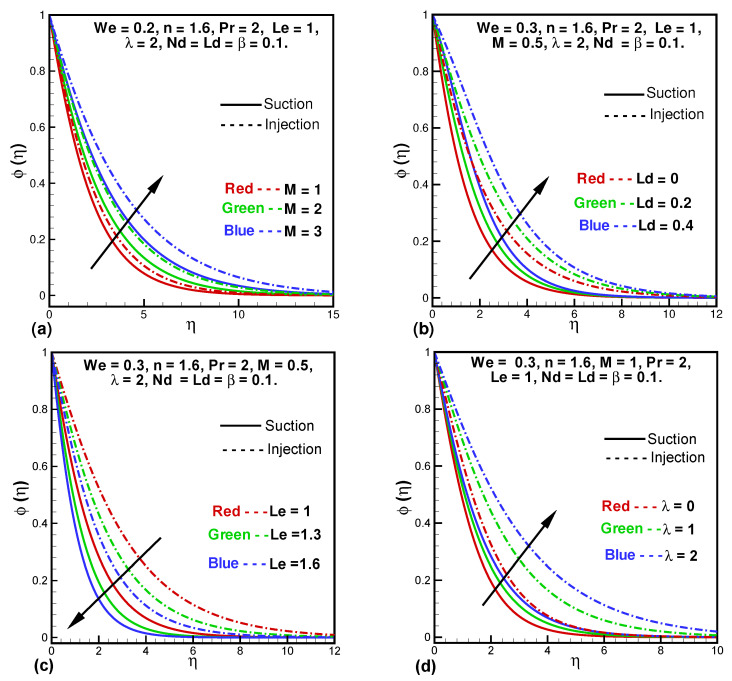
Variationsin concentration profile against *M* (**a**), Ld (**b**), Le (**c**), and λ (**d**).

**Figure 5 entropy-24-00377-f005:**
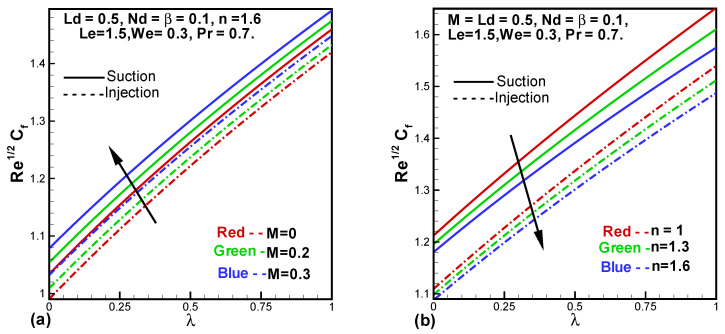
Behavior of skin friction factor against *M* and λ (**a**), *n* and λ (**b**).

**Figure 6 entropy-24-00377-f006:**
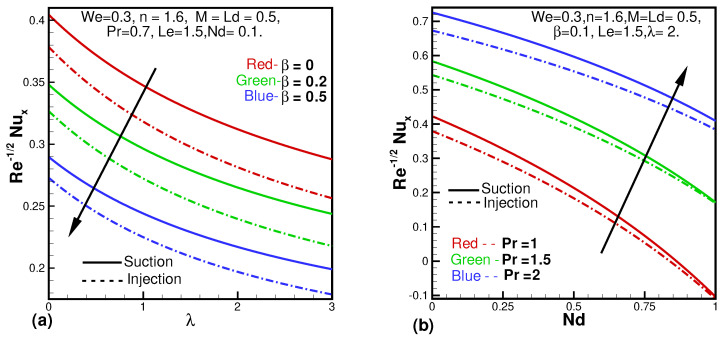
Behavior of Nusselt number against β and λ (**a**), Pr and Nd (**b**).

**Figure 7 entropy-24-00377-f007:**
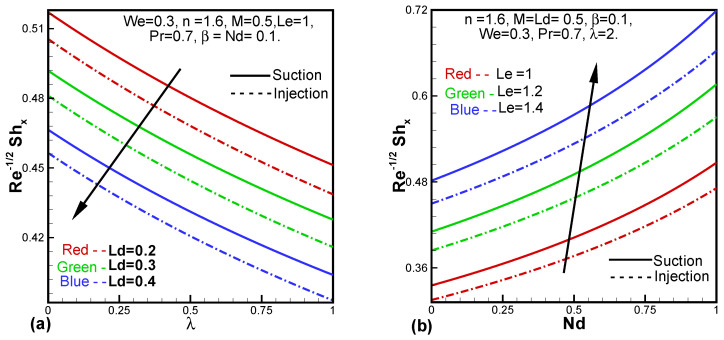
Behavior of the Sherwood number against Ld and λ (**a**), Le and Nd (**b**).

**Table 1 entropy-24-00377-t001:** Comparison of the skin friction for various values of *M*, and Pr=Nd=Ld=Le=β=λ=We=n=0.

Values of *M*	Khan et al. [30]	Current Results
0	1	1
0.5	−1.1181	−1.1181
1	−1.4141	−1.4142

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
