# Peer review of "MHD Double-Diffusive Carreau Fluid Flow through a Porous Medium with Variable Thermal Conductivity and Suction/Injection"

_entropy, 2022, doi:10.3390/e24030377_

Round 1

Reviewer 1 Report

Non-Newtonian fluid had covered a large range of fields either in industrial or in biomedical applications, and it is important to explore its flow and heat transfer characteristics. The current work was to investigate the double diffusive magnetohydrodynamics (MHD) Carreau fluid in a porous media. Generally, the topics seems interesting, however, fetal problems exist. I do not recommend its publication.

  1. The main content has little relationship with the scope of Journal Entropy.
  2. There seems little innovation.
  3. The manuscript was not well written.

Reviewer 2 Report

The double diffusive MHD carreau fluid in a porous media is studied by presenting velocity, concentration,
temperature, skin friction coefficient , Nu and Sh for different parameters. However, the manuscript
is loosely organized, especially the writing of introduction needs to be improved. To be published, the following 
revisions have to be made
1. Language needs to be polished, there are so much typos, for instance
 In Page 1, "The results shows...".
In Page 4, "Figures 1(a-d) shows...","the fluid is decreases...".
"we taken..." in Paragraph 1 of Section 3.
"Figure 6(a) demonstrate..." in Page 5
Captions of Figure 2.
“T, T_w,and T_inf represents...”in Page 3
2. The manuscript should prepared according to the Template file. For instance 
(Keywords should be separated with semicolon. line number should be presented)
3. Why the case of power-law index n<1 is not considered?
4. What does symbol a represent?
5. Abstract should be concise and informative.
6. Application of the considered porous medium model should be presented in Paragraph 2 in Page2.
And how does the present manuscript advances the study of fluid flow passing through a stretching sheet>
7. The sketch of the considered fluid flow should be presented.
8. In figures, I suppose the illustration "Red--" means both the red  '-.-.'line and red'-'line in graphs, while
which are confusing, because which might be misunderstood as the illustration of red '-.-.' line.
9. The physical interpretation of important governing parameters should be presented.
10. The cases of suction and injection are considered, however, why is there no discussion and conclusion 
regarding the difference between suction and injection for different parameters, or what is 
the significance of the consideration of two cases?
11. There is no illustration on Table 1.
12. Please provide relevant  information on the application of similarity transformation.

Reviewer 3 Report

Analysis of double diffusion and thermal conductivity on Carreau fluid through a permeable stretching sheet

The present study is focusing on the double diffusive magnetohydrodynamics (MHD) Carreau fluid in a porous media. Thermal conductivity is considered a varying quantity. Similarity transformations are utilized to transform the equations governing the Carreau fluid to dimensionless non-linear ordinary differential equations.

  • Correct the typo error in equation -6(Similarity variable /eta)
  • Why Weissenberg number contains variable x? since Similarity analysis used to solve governing equations. Justification  is necessary
  • Why this method is used for the solution, gives the advantages.
  • How to choose the values of parameter, explain?
  • Why the Prandtl number Pr=2? Authors considered Carreau fluid. So by choosing suitable value corresponding to Carreau fluid, re draw all the graphs.
  • Check the dimension of Dufour solutal paramete
  • An updated and complete literature review should be conducted and should appear as part of the Introduction. In this regard I suggest following papers on stretching sheet… Influence of Thermophoretic Particle Deposition on the 3D Flow of Sodium Alginate-Based Casson Nanofluid over a Stretching Sheet, The Impact of Cattaneo–Christov Double Diffusion on Oldroyd-B Fluid Flow over a Stretching Sheet with Thermophoretic Particle Deposition and Relaxation Chemical Reaction, Analysis of transient thermal distribution in a convective–radiative moving rod using two-dimensional differential transform method with multivariate pade approximant, Effect of magnetohydrodynamics on heat transfer behaviour of a non-Newtonian fluid flow over a stretching sheet under local thermal non-equilibrium condition, Comprehensive study of thermophoretic diffusion deposition velocity effect on heat and mass transfer of ferromagnetic fluid flow along a stretching cylinder, Impact of magnetic dipole on thermophoretic particle deposition in the flow of Maxwell fluid over a stretching sheet

Reviewer 4 Report

This manuscript numerically investigates a double diffusive Carreau non-Newtonian fluid with varying thermal conductivity in a porous media. In my opinion the topic is interesting and the article may be useful for future researchers in the topic. I have only the following comments for authors:

  • The introduction is well organized. However, authors should emphasize on what makes present investigation different from previous research. Particularly, authors should better describe differences with previous ref. [25].
  • In some parts of the article, authors should unabbreviated some magnitudes (it would ease the reading). Likewise, a nomenclature table could be of great help for the reader.
  • Problem formulation looks somewhat incomplete. In this section, authors should not limit to present used equations but also include some discussion regarding considered assumptions. A schematics (or figure) could be of great help for better understanding the problem.
  • More information regarding investigated conditions is also necessary.
  • Even if graphs are, in general, well presented, captions could be more informative.
  • Section 4 merely describes results. More contrast with previous research would be necessary.

Round 2

Reviewer 1 Report

In the current version of manuscript, the author make careful revision. The quality of the paper seems to be improved. It is suggested that the author could discuss the mechanism or explanation of the tendency of results, ranging from Fig. 2 to Fig.7. 

Author Response

  1. In the current version of manuscript, the author make careful revision. The quality of the paper seems to be improved. It is suggested that the author could discuss the mechanism or explanation of the tendency of results, ranging from Fig. 2 to Fig.7. 

            Response: The suggestions of the reviewer 1 has been incorporated in the manuscript. It is also  highlighted in the revised version of the paper.

Reviewer 2 Report

The manuscript can be accepted in the present version.

Author Response

Reply submitted earlier is accepted.

Reviewer 3 Report

Paper can be accepted in the present form

Author Response

Reply submitted earlier is accepted.

Reviewer 4 Report

Authors have performed the changes suggested by this reviewer. I only have the following minor remark:

- Nomenclature table should be organized following a logical criterion (alphabetical order, for instance). Pay notice that “thermal conductivity” appears twice. When possible, indicate the units.

Author Response

  1. Nomenclature table should be organized following a logical criterion (alphabetical order, for instance). Pay notice that “thermal conductivity” appears twice. When possible, indicate the units.

            Response: As per referee comments, nomenclature has been rechecked/revised and the suggestions are incorporated in the manuscript.